# EMPIRICALLY VERIFYING HYPOTHESES USING REINFORCEMENT LEARNING

## ABSTRACT

This paper formulates hypothesis verification as an RL problem. Specifically, we aim to build an agent that, given a hypothesis about the dynamics of the world, can take actions to generate observations which can help predict whether the hypothesis is true or false. Existing RL algorithms fail to solve this task, even for simple environments. We formally define this problem, and develop environments to test different algorithms' performance. We analyze methods which are given additional pre-training rewards and find that the most effective of these is one that exploits the underlying structure of many hypotheses, factorizing them as {*pre-condition*, *action sequence*, *post-condition*} triplets. By leveraging this structure we show that RL agents are able to succeed. Furthermore, subsequent fine-tuning of the policies allows the agent to correctly verify hypotheses not amenable to this factorization.

## 1 INTRODUCTION

Empirical research on early learning Gopnik (2012); Kushnir & Gopnik (2005) shows that infants build an understanding of the world by constantly formulating hypotheses about how some physical aspect of the world might work and then proving or disproving them through deliberate play. Through this process the child builds up a consistent causal understanding of the world. This contrasts with manner in which current ML systems operate. Both traditional i.i.d and interactive learning settings use a single user-specified objective function that codifies a high-level task, and the optimization routine finds the set of parameters (weights) which maximizes performance on the task. The learned representation (knowledge of how the world works) is embedded in the weights of the model - which makes it harder to inspect, hypothesize or even enforce domain constraints that might exist. On the other hand, hypothesis generation and testing is a process explored in classical approaches to AI Brachman & Levesque (2004). In this paper we take a modest step towards the classical AI problem of building agents capable of testing hypotheses about its world using modern ML approaches.

The problem we address is illustrated in Figure 1. Agents are placed in a world which has several interactive elements. They are provided with a hypothesis (an "action sentence" Pearl (2009)) about the underlying mechanics of the world via a text string (e.g. "$\mathcal{A}$ will be true if we do $\mathcal{B}$"). The task is to determine if the hypothesis is true or not. This problem cannot be solved without *interaction* with a dynamic world (comparing the state before and after taking action $\mathcal{B}$).

A key novelty in our work is formulating the task in a manner that permits the application of modern RL methods, allowing raw state observations to be used rather than abstract Boolean expressions of events. To do this, we use a model composed of two different deep parametric functions which are learned through interaction: (i) a policy that generates observations relevant to verification of the hypothesis and (ii) a prediction function which uses the observations to predict whether it is true.

We first show that agents trained end-to-end using deep RL cannot learn policies that can generate observations to verify the hypothesis. To remedy this, we exploit the underlying structure of hypotheses – they can often be formulated as a triplet of a pre-condition ($\mathcal{P}$), an action sequence (collectively $\mathcal{B}$), and a post-condition ($\mathcal{A}$) that is causally related to the pre-condition and actions. Using this structure, we can seed our action policy to learn behaviors which alter the truth of the pre-condition and post-condition. This allows agents to learn policies that can generate meaningful observations for training the prediction function. We further demonstrate that these policies can be adapted to learn to verify more general hypotheses that do not necessarily fit into the triplet structure. Our experiments

Figure 1: Example "crafting" world. The agent verifies a hypothesis (provided a text). Acting according a learned policy, the agent manipulates the observation to one that allows a learned predictor to determine if the hypothesis is true. The learning of policy and predictor is aided by a pretraining phase, during which an intermediate reward signal is provided by utilizing hypotheses that factor into {*pre-condition state*, *action sequence*, *post-condition state*}.

show that this approach outperforms naive RL and several flavors of intrinsic motivation designed to encourage the agent to interact with the objects of interest.

## 2 RELATED WORK

**Knowledge representation and reasoning (KRR)** Brachman & Levesque (2004) is a central theme of traditional AI. Commonsense reasoning Davis (1990); Davis & Marcus (2015); Liu & Singh (2004) approaches, e.g. CYC Lenat (1995), codify everyday knowledge into a schema that permits inference and question answering. However, the underlying operations are logic-based and occur purely within the structured representation, having no mechanism for interaction with an external world. Expert systems Giarratano & Riley (1998) instead focus on narrow domains of knowledge, but are similarly self-contained. Logic-based planning methods Fikes & Nilsson (1971); Colaco & Sridharan (2015) generate abstract plans that could be regarded as action sequences for an agent. By contrast, our approach is statistical in nature, relying on Reinforcement Learning (RL) to guide the agent.

Our approach builds on the recent interest Mao et al. (2019); Garcez et al. (2012) in neural-symbolic approaches that combine neural networks with symbolic representations. In particular, some recent works Zhang & Stone (2015); Lu et al. (2018) have attempted to combine RL with KRR, for tasks such as navigation and dialogue. These take the world dynamics learned by RL and make them usable in declarative form within the knowledge base, which is then used to improve the underlying RL policy. In contrast, in our approach, the role of RL is to verify a formal statement about the world. Our work also shares some similarity with Konidaris et al. (2018), where ML methods are used to learn mappings from world states to representations a planner can use.

**Causality and RL:** There are now extensive and sophisticated formalizations of (statistical) causality Pearl (2009). These provide a framework for an agent to draw conclusions about its world, and verify hypothesis as in this work. This is the approach taken in Dasgupta et al. (2019), where RL is used to train an agent that operates directly on a causal Bayesian network (CBN) in order to predict the results of interventions on the values on its nodes. In contrast, the approach in this work is to sidestep this formalization with the hope of training agents who test hypotheses without building explicit CBNs. Unlike Dasgupta et al. (2019), our agents intervene on the actual world (where interventions may take many actions), rather than the abstract CBN. Nevertheless, we find that it is necessary to add inductive bias to the training of the agent; here we use the pretraining on $(\mathcal{P}, \mathcal{B}, \mathcal{A})$ triplets. These approaches are complementary; one could combine explicit generation and analysis of CBNs as an abstract representation of an environment with our training protocols.

Our work is thus most similar to Denil et al. (2016), which uses RL directly on the world, and the agent gets reward for answering questions that require experimentation. However, in that work (and in Dasgupta et al. (2019)), the "question" in each world is the same; and thus while learning to interact led to higher answer accuracy, random experimental policies could still find correct answers. On the other hand, in this work, the space of questions possible for any given world is combinatorial, and random experimentation (and indeed vanilla RL) is insufficient to answer questions.

**Cognitive development:** Empirical research on early learning Gopnik (2012); Kushnir & Gopnik (2005) shows infants build an understanding of the world in ways that parallel the scientific process: constantly formulating hypotheses about how some physical aspect of the world might work and then proving or disproving them through deliberate play. Through this process the child builds up an abstract consistent causal understanding of the world. Violations of this understanding elicit measurable surprise Spelke et al. (1992).

**Automated Knowledge Base completion:** This work is also related to knowledge base completion Fader et al. (2011); Bordes et al. (2013); Suchanek et al. (2007), especially as formulated in Riedel et al. (2013). Instead of using facts in the knowledge base or a text corpus to predict edges, here the agent acts in a world and observes the results of its actions. This recalls Mitchell et al. (2018), where the system verifies facts it had hypothesized by searching for corroboration in the corpus.

**Automation of the scientific process:** has been tried in several domains. Robotic exploration of chemical reactivity was demonstrated Granda et al. (2018) using ML techniques. King et al. (2009) developed a robot scientist that explored geonomics hypotheses about yeast and experimentally tested them using laboratory automation. In biochemistry Vanlier et al. (2014) used Bayesian methods for optimal experiment design. More generally, the Automated Statistician project Steinruecken et al. (2019) uses a Bayesian approach to reason about different hypotheses for explaining the data, with the aim of creating interpretable knowledge.

**Embodied Question and Answering:** The problem studied in this paper is closely related to the embodied visual question-answering problem in Das et al. (2018). Indeed, our basic formulation is a particular case of the most general formulation of embodied QA, as the agent is rewarded for successfully answering questions about the world that require interaction. However, the form of the questions is different than those considered in that work, as they may require drawing a conclusion about the *dynamics* of the world, rather than a static property. Even the questions about static properties we are interested in have a different flavor, as they encode rules, rather than statements about the current configuration. Our approach is built around hypothesis-conclusion structure special to these questions. There is also a large body of work on visual QA Kafle & Kanan (2017); Wu et al. (2016a) and text-based QA Rajpurkar et al. (2018). From this, most relevant to our work is Wu et al. (2016b) who use a structured knowledge base to augment standard QA techniques.

## 3 THE HYPOTHESIS VERIFICATION PROBLEM

An agent is spawned in a world sampled from a distribution over possible worlds. In the case of "Crafting", shown in Figure 1, there are items lying around that the agent can pick up and combine using a "craft" action. The exact dynamics change for every newly instantiated world; so in one world, taking a craft action with a stick might produce a torch, and in another, it might produce a pickaxe. At the start of each episode, the agent is given a hypothesis about the world, such as the one shown at the top of Figure 1. The agent gets a reward when it correctly answers if that hypothesis is true or false. Because the dynamics and rules change each episode, the agent must learn to interact with the world in order to decide if the hypothesis is true. In Figure 1 the agent picks up the stick and does a craft action to see that a torch is created. It then has enough information to decide the hypothesis is true, and the agent receives reward for verifying the hypothesis correctly.

In this work, we will structure our hypotheses using templated language. One could imagine using more expansive formal symbolic systems (e.g. first order logic), or alternatively, using natural language descriptions of the hypotheses. The former might allow interfacing with symbolic solvers or otherwise using combinatorial approaches; whereas the latter would allow scaling annotation to untrained humans. We choose templated language because it is simple, and sufficient for the environments on which we test, which are already challenging for standard RL. Moreover, in our view it is a good starting point for further work that would use either more sophisticated formal representations or more natural language representations.

**Formal Definition** We define a world as a set of states and actions with Markovian dynamics (an MDP without reward). We define an environment $\mathcal{E}$ as a distribution over a set of worlds $\mathcal{W}$ and hypotheses $\mathcal{H}$. A world $W \in \mathcal{W}$ is specified by rules $L_W$ describing the dynamics of the world. We define this reward-less MDP of one specific world $W$ as $MDP_W = \{S_W, A_W, T_W\}$ where state space $S_W$ includes the position and state of objects in the world (e.g. the placement of the agents and the object), $A_W$ is the action space, and $T_W$ is the transition function. Note that $T_W$ depends on $L_W$, the rules of this specific world. Actions have different consequences depending on $L_W$.

Now $\mathcal{E}$ is an episodic POMDP where each episode consists of sampling[1] a $W$ and $h$. ($G$ is a ground-truth function that takes in the hypothesis $h$ and world $\mathcal{W}$ and outputs {true, false}. In this work, hypotheses are generated via templated language and their truth function $G$ depends on $W$, more specifically $L_W$. The episode ends when the agent executes either the true or false action.

---

[1]See Appendix B for details on sampling procedures

Given a world $W$ and hypothesis $h$, an agent gets reward:

$$R_{Hyp} = \begin{cases} +1 & a = G(h, W) \\ -1 & a = \neg G(h, W) \\ 0 & otherwise \end{cases}$$

The observation in this POMDP is $o = (s_W, h)$, the state from the world $W$ plus the hypothesis. The state is $s = (s_W, h, L_W)$. This includes the rule $L_W$ which is not visible in the observation. The action space is just $A_W \cup \{\text{true}, \text{false}\}$ for any $W$ (they are the same for a given environment); and $T = T_W$. Note that the transition function $T$ depends on the (hidden) $L_W$. The goal of hypothesis verification is now to discover the truth of $h$, which depends on $L_W$.

## 4 METHODS

**RL baseline** Given the formulation of the hypothesis verification problem as a POMDP, we could try to solve it using an RL agent with $a = \pi(O_t, h)$, where $O_t = \{o_t, o_{t-1}, \ldots o_{t-K}\}$. Here $o$ is an observation of the current world, $K$ is a history window size, and $h$ is the hypothesis for the current world. We found that standard RL agents struggle to solve this problem.

To make the problem easier to solve, we can augment the reward with direct supervision of a prediction network $f(O_t, h)$ which takes in the last $K$ observed observations and the hypothesis and predicts whether or not the hypothesis is true. The policy network now, instead of taking true or false actions, takes a special stop action which is replaced by true or false based on the prediction of $f$.

Even with the augmented supervision, as shown in Fig. 4, an RL baseline is not able to solve the task. In order determine whether a hypothesis is true, the agent needs to take the correct sequence of actions related to the hypothesis. But in order to know that a particular sequence of actions was the right one, it needs to be able to correctly predict the hypothesis. Guessing with no information gives a zero average reward, and despite the supervision on the output of the predictor, the predictor does not see state transitions that allow it to learn.

**Motivation to interact the world** Given that naive RL and supervised learning is not up to our task, how can we encourage policies to explore the world? The first thing we might think of would be to initially reward agents for "exploring" the world. This can be done by encouraging the agent to explore the world to minimize prediction error Pathak et al. (2017). We more directly incentivize this kind of exploration by encouraging our agents to interact with the elements of the world we know are related to our hypotheses. We test four variants of an "intrinsic" reward to see if other pretraining schemes might perform equally well. We show results on the gridworld domains using 4 different intrinstic forms of pretraining: (i) change any item state in the world; receive reward at end of episode. (ii) change any item referenced in the hypothesis; receive reward at end of episode. (iii) change any item state in the world; receive reward instantaneously. (iv) change any item referenced in the hypothesis; receive reward instantaneously. Here, "item" means any object that is not the agent (including crafting items, switches, pushblocks, etc.). See Appendix H for additional details.

Note our baselines more directly reward the agents for exploring the specific subject of the hypotheses compared to Pathak et al. (2017). Despite this, these methods still have relatively poor performance on our problem (see Section 6). Is there a way we can more directly train our policies to explore the cause and effect relationship of our hypotheses?

**Pretraining using Triplet Hypotheses** In light of the difficulties directly training an RL model using terminal reward and hypothesis prediction supervision, we take advantage of the fact that many causal statements about the world have the form: $(\textit{pre-condition}, \textit{action sequence}) \implies \textit{post-condition}$

We define this formally in Appendix B.2, but informally this means that when the state meets a "precondition" and an "action sequence" is taken, this will result in the state meeting a "post-condition." In Figure 1 the pre-condition is having a stick and being at craftingtable, the action sequence is craft, and the post-condition is that a torch is made.

This structure can be converted into a reward function that can be used to pretrain the agent policy $\pi$. The idea is to reward the agent for taking actions which alter the truth of the pre-condition and post-condition (i.e. changing the world state so that pre/post-conditions are met or not). If it matches the pre-condition state and takes the action, if the statement is true, the post-condition should toggle from false to true in the world. Similarly, if post-condition changes but the pre-condition did not

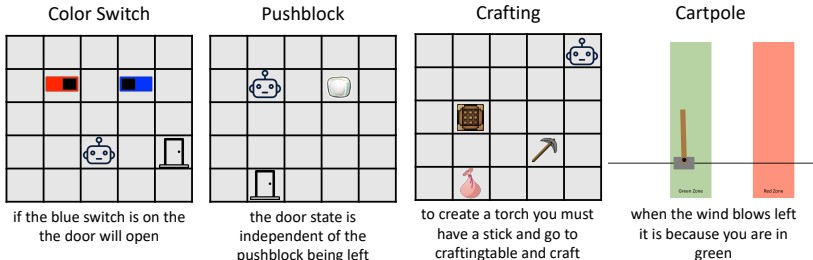

Figure 2: Examples of the four environments used in our experiements: ColorSwitch, Pushblock, Crafting and Cartpole.

change, the statement must be false. This can be formalized in the following reward function to encourage the agent to toggle pre-condition and post-condition states:

$$
R_{\text{pre}} = \begin{cases} +C & a = \text{stop } \& \\ & \text{pre-condition changed in last } K \text{ steps} \\ 0 & otherwise \end{cases}
$$

$$
R_{\text{pre+post}} = \begin{cases} +C & a = \text{stop } \& \text{ post+pre-condition} \\ & \text{changed in last } K \text{ steps} \\ 0 & otherwise \end{cases}
$$

This encourages the policy $\pi$ to change the pre-condition and post-conditions (via pre-condition) in the last $K$ steps, so that a predictor looking at the last $K$ observations will be able to deduce the truth value of the hypothesis. More generally, training with this reward function forces the policy network to ground text concepts (e.g. the text "stick" means [object_stick]) and also captures the causal rules within the world. Consequently, following pretraining, the policy network can then be fine-tuned using the original reward function $R_{\text{Hyp}}$. Since the policy network is no longer random, a robust prediction network $f$ can also be learned. While not all hypotheses fit into the triplet format, we show in the experiments that the knowledge captured by the policy and prediction networks during this phase of training can generalize to less structured hypotheses. We have a set of hypotheses for each world that contains only these triplet-structured hypotheses. We use this set for pretraining.

**Training using Triplet hypothesis** After pretraining the policy $\pi$, we further train $\pi$, as well as the prediction network $f$ using the same set of triplet hypotheses, but now using $R_{\text{Hyp}}$ instead of $R_{\text{pre}}$ or $R_{\text{pre+post}}$. Two variants are explored: (i) "fixed" – keep $\pi$ fixed but train the prediction network $f$ and (ii) "finetune" – finetune $\pi$ and train $f$. Performance in this phase is used to select promising models for subsequent stages of training. Specifically, runs achieving <90% accuracy (see the appendix for alternate cutoff) are eliminated.

**Adaptation to non-triplet hypotheses** Next, we want to show that we can adapt these networks to hypotheses other than those that fall neatly into the triplet structure. To adapt to the larger set of hypotheses, we start with the networks trained previously on triplet templates. During this training stage, the triplet-form constraint is relaxed and training proceeds with both triplet and non-triplet hypotheses (see Sec. 5), using an even split between the two types.

## 5 EVALUATION ENVIRONMENTS

In this section we describe our environments. The environments $\mathcal{E}$ are designed so that the prior probability $p(h = true) = 0.5$ and the initial observation $o_0$ does not contain information about $h$.

**Environments** We created four different environments for hypothesis verification. ColorSwitch, Pushblock and Crafting are all gridworld-based environments. A fourth enviornment is created by adapting the standard Cartpole task to include interactive elements. See Fig. 2.

**ColorSwitch**: The agent is placed in a world with one or more color switches (randomly on or off) and a door (open or closed). The agent can move and toggle the switch positions. One switches, when in the correct position (on/off) will cause the door to open. The other switches have no effect. Hypotheses relate to the color and position of switches and how that opens or closes the door.

**Pushblock**: The agent is placed in a world with a block which can be pushed by the agent, and a door. The agent can push on the block. The door opens when the block is in a particular part of the

grid: "up" – top two rows, "down" – bottom two rows, "left" – leftmost two rows, "right" – rightmost two rows. The hypotheses relate to the position of the pushblock and how that affects the door.

**Crafting**: The agent is placed in a world with crafting rules similar to that of the popular Minecraft game. The agent is spawned along with a number of crafting items, and a crafting location. The agent is able to move, pick up items into its inventory and use the crafting location using special crafting actions. There is some true "recipe" which produces some new item in the agent's inventory.

**Cartpole**: This is the standard cartpole problem where a pole is attached by an un-actuated joint to a cart. The actions are "left" and "right" and if the pole falls, the episode ends. In our modification, there are "zones" (an interval on the x axis) where the physical laws change by either changing the gravity constant, or applying a "wind force" blowing the cart in one direction. Like in ColorSwitch, the zones are specified by color. Typically one color zone has an effect and the other is a decoy zone that has no effect. The hypotheses relate to which zones correspond to changes to the physics.

In the grid world environments, items are randomly generated in a 5 by 5 grid. The world observation is given by a 1-hot vector for each grid location and inventory. The hypothesis is encoded as sequence of tokens. In Cartpole the state is the standard dynamics as well as a 1-hot vector specifying the location and color of the zones.

**Hypothesis Construction** We now describe how the hypotheses for each world in each environment are automatically generated via templates. Three different varieties are considered: (i) triplet hypotheses, (ii) general templates and (iii) special case templates. (See Appendix B.3 for all of the possible templates for an environment and further details).

**Triplet hypotheses**: Here hypotheses take the form of an logical statement: (pre-condition, action sequence) $\implies$ post-condition. When the pre-condition is true, and the action sequence is performed, the post-condition becomes true. To generate triplet hypotheses, we: (i) randomly select a pre-condition template from a set list; (ii) randomly select an action template; (iii) randomly select a post-condition template; and (iv) fill in any entities. In our example from Fig. 1 this would be ("when you are at crafting table and you have stick"; "and then you craft"; "then torch is made").

**General templates**: Instead of drawing a template from the triplet form, a single template for the hypothesis is drawn and the values populated. For instance, in Pushblock, a template might be "the door can only be opened when the pushblock is PUSHBLOCK_POSITION" and then "left" would be drawn for PUSHBLOCK_POSITION. These templates are more general than the triplet ones in that they have no explicit (pre-condition, action sequence and post-condition) structure.

**Special cases**: We also use more difficult and general hypothesis templates. These cannot be neatly fit into a triplet format by rewording, and may not fully describe the rules of the world. Some examples of these harder templates are: (i) Negating effects (e.g. "door is not open"); (ii) Negating conditions (e.g. "switch is not on"); and independence (e.g. "door independent of blue switch").

## 6 EXPERIMENTS

Figure 3 shows results from learning with pretraining rewards $R_{\text{pre+post}}$ and $R_{\text{post}}$. There is relatively little variance, with all runs achieving near the theoretical maximal rewards[2] For Crafting and Cartpole, $R_{\text{pre+post}}$ is not always achievable if true and distractor items are far away from each other. See Appendix G.1 for further discussion. [3]

In Figure 4, we show the results on non-triplet adaptation (Sec. 3). As discussed in Section 5, this stage includes the more difficult, non-triplet templates not seen during pretraining or during triplet hypothesis training. We also break down the final hypothesis prediction accuracy for the triplet methods in Table 1. This allows us to see whether triplet methods were able to adapt to non-triplet hypotheses. All network architectures for the policy and prediction networks, as well as hyper-parameters are the same for all methods.

**RL baseline** Figure 4 shows the RL baseline at chance-level performance, the only exception being CartPole were it achieves $\sim 60\%$, versus the $\sim 90\%$ of triplet methods. Note that this is the RL

---

[2]For Pushblock, sometimes the block can be stuck against a wall, so not all worlds are solvable

[3]For gridworld environments we show variance runs on 25 seeds and 5 for Cartpole. We provide further hyperparameters, and training details in Appendix C and D and network architecture and details in Appendix D

baseline *with* the supervised predictor, as discussed in Section 3. This poor performance is a result of training both policy and predictor from scratch; see Figure 6.

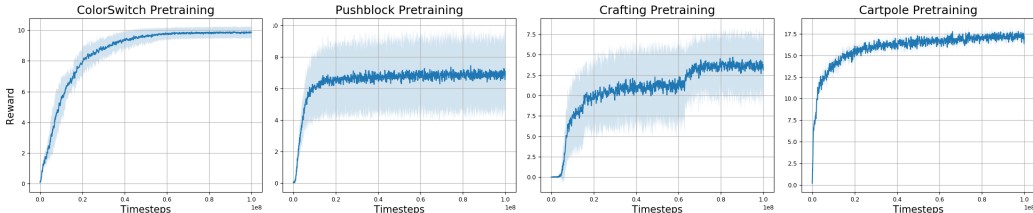

Figure 3: Pretraining reward ($R_{\text{pre}} + R_{\text{pre+post}}$) on our four environments.

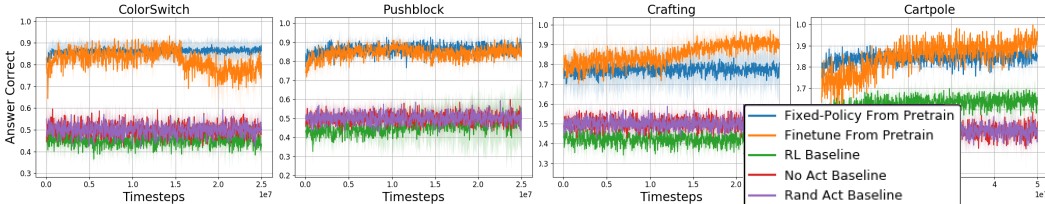

Figure 4: Hypothesis prediction accuracy on *both* triplet *and* non-triplet hypotheses (1-1 ratio) for the ColorSwitch, Pushblock, Crafting and Cartpole environments, using $R_{\text{Hyp}}$ reward for training.

**Other baselines** We also include two other simple baselines "no act" and "random act." The "no act" baseline simply takes the stop action at $t = 0$, forcing the prediction network to give an answer from just the first observation. This fails because the agent needs to take actions in the world to be able to predict the hypothesis accurately. This also confirms that we do not accidentally leak the ground truth of $h$ into the initial observation. For "random act", a random policy is used (i.e. uniform across actions). This fails as random actions are extremely unlikely to verify the hypothesis.

**Discussion** From these results, we can clearly see that naive RL and other baselines cannot efficiently solve hypothesis verification tasks. When we use triplet pretraining we use the insight that hypotheses often have a clear causal structure that can be exploited when they are formed as "triplet hypotheses." Not all hypotheses fall neatly into this form, and we may not have this form for all hypotheses. But if we have some that fit this form, we can gain a foothold that lets us make progress on this problem, and then later adapt them to other hypotheses. From Figure 4 we can see that this pretraining, triplet training and adaptation works. We also show in Section 6 that other plausible pretraining rewards fail to achieve the same results as our method.

Looking at the different environments, we see that in Pushblock and ColorSwitch, even with the policy learned from the triplet pre/post reward, the agent is able to generalize and perform well on hypotheses not seen in the pretraining phase as we can see in Table 1. Because the finetuning step adds additional variance to training, there is a small, but non-trivial drop in average performance. This drop is less extreme when comparing max values (see Appendix G.3).

Table 1: Average Hypothesis Prediction scores, broken down by triplet (pretrained) and non-triplet (not seen in pretraining)

|  | Method | Overall | Triplet Accuracy | Non-triplet Accuracy |
|---|---|---|---|---|
| **ColorSwitch** | Fixed Policy | 86.6% | 91.1% | 82.1% |
| | Finetuned Policy | 77.5% | 79.7% | 75.4% |
| **Pushblock** | Fixed Policy | 86.9% | 87.9% | 85.9% |
| | Finetuned Policy | 85.6% | 86.3% | 84.8% |
| **Crafting** | Fixed Policy | 77.3% | 92.8% | 61.8% |
| | Finetuned Policy | 90.7% | 98.4% | 83.0% |
| **Cartpole** | Fixed Policy | 84.2% | 92.0% | 76.3% |
| | Finetuned Policy | 92.5% | 93.4% | 91.6% |

In Crafting and Cartpole on the other hand, to do well on the unseen templates, the policy also needs to be fine-tuned. This tells us that when we do have to generalize to unseen hypotheses (especially non-triplet hypotheses), adapting the policy as well as the prediction network is necessary. Recall

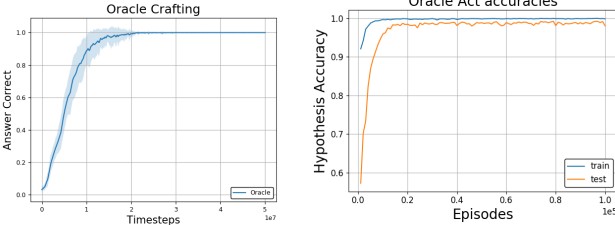

Figure 5: Hypothesis accuracies with intrinsic pretraining versus triplet pretraining (purple).

that we test very different hypotheses such as as negations and "independence" hypotheses not see in triplets (see suplementary). We see from Table 1 that indeed, finetuned policies greatly outperform the fixed policies on the non-triplet templates.

We further break down these results by specific templates, entities mentioned (e.g. colors) and whether the given template was true or false. We find for instance that for most environments it is harder to determine a hypothesis is false than true. See Appendix F for the full analysis.

**Alternate forms of pretraining**

In Figure 5 we show the accuracies on the final hypothesis verification task for both triplet and non-triplet hypotheses, using the four intrinsic pretraining methods discussed in Setion 3. We also plot the final average accuracy obtained by triplet methods from Figure 4. For the intrinsic pretrained policies the best run is shown to show the best-possible case of the alternative methods. For Crafting the dense intrinsic pretraining works about as well as triplet pretraining. This can be explained by the fact that this particular form intrinsic pretraining directly rewards the agent for doing many of the operations in the actual Crafting task, i.e. picking up objects and crafting objects. However, averaging across the three environments, all the intrinsic pretraining methods do worse.

**Oracle ablations** We also show two oracle analysis in the Crafting environment. In the first, we provide an "oracle" hypothesis predictor which will output the ground truth of the hypothesis if it is inferable from the last $K$ frames, and test whether we can learn a policy directly using reward $R_{\mathsf{Hyp}}$. Similarly, we also train a hypothesis prediction network with observations from an oracle policy network (observations from which the ground truth of the hypothesis is always inferable).

Figure 6: (Left) Results on training an RL using $R_{\mathsf{Hyp}}$ with oracle predictor on the Crafting environment. (Right) Results training just hypothesis prediction on oracle policy.

Figure 6 (left) shows the prediction oracle agent quickly converging. From this we surmise that if we can predict the hypothesis already, learning an optimal policy is relatively straightforward. Similarly, (right) show that with the correct observations from an oracle policy, the hypothesis predictor is able to quickly converge as well. This shows the joint nature of the problem (i.e. learning both policy and predictor) is what makes it challenging for the RL baseline.

## 7 DISCUSSION

In this work, we propose a tractable formulation of the problem of training agents that can interact with a world to test hypotheses. We show that generic RL techniques struggle with the problem, but we show that methods exploiting the structure of hypotheses are able to successfully complete the problem. We hope that our formulation and analysis serve as a starting point for researchers to explore the problem of hypotheses verification and further on how we can build towards agents that can ask and answer questions about the world.

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

## A    APPENDICES

Our Appendices include additional experiments, further details on environments and methods, standard hyper-parameters and an exhaustive list of the hypothis templates used in this work.

In Appendix B, in B.1 we further detail how we randomly construct environment rules and hypotheses. In B.2 we provide a fuller definition of our notion of "triplets". And in B.3 we exhaustively list our templates and named entities.

In Appendix C we provide additional training details including training hyperparameters.

In Appendix D we provide details and diagrams of our neural networks, a brief ablation analysis, and network hyperparameters.

In Appendix E we show results of the triplet pretraining methods on held-out test hypotheses.

In AppendixF we show the performance of the triplet methods on different crosstabs of the hypotheses and rules, including results on each individual hypothesis.

In Appendix G we provide additional details of the triplet pretrain methods and justification of some parameter choices and results.

Finally, in Appendix we describe the intrinsic pretraining method in a bit more detail and show training curves for the pre-training objectives

## B    TEMPLATES

### B.1    WORLD AND HYPOTHESIS CONSTRUCTION

Returning again to our notation from the main paper, the environment at each spawn needs to construct a world $W$, and a hypothesis $h$ that is either true or false in the world. $W$ in particular describes the rules about how the environment works (i.e. which switch opens the door) which in our case can precisely be describe by a hypothesis. So given a true hypothesis, we can exactly describe the rules of the world. Therefore, in order to create an instance of a possible $W$, we can instead draw a true hypothesis about the world at random. From the hypothesis, we can then construct the rules the determine how objects in the world behave. Note that there are couple exceptions to this for our harder hypotheses, where the hypothesis can be true but only partially describes all the rules of $W$. For these cases, we draw yet another template which is consistent with the hypothesis and use that to construct the rules, such as deciding which color switch really opens the door.

Because we have to randomly give either a true or false hypothesis, we also need to be able to generate a false hypothesis for the world. So for every instance, we also draw a random false hypothesis. Now, given a true and false hypothesis, we can fully generate the world and all the items that appear in either statement. So for instance, if the true hypothesis mentions a green switch and the false one mentions a blue switch, we generate both a green and blue switch. Then, we can set the rules such that the right thing happens. So in this example, switching the green switch opens the door and the blue switch does nothing.

The final step is then to randomly choose either the true or false statement as the "visible" hypothesis which is passed to our agent to verify. Because we generate the world and spawn the items before we make this choice, we ensure that we do not accidentally give away the truth of the hypothesis based on what items spawned.

Our process for generating a new spawn of environment can thus be summarized as follows:

1. We randomly generate a true hypothesis
2. We randomly generate a false hypothesis
3. We construct a ruleset from the true hypothesis
4. We spawn the agent and the items in the world described in both the true and false hypothesis
5. We randomly choose either the true or false hypothesis as the "visible" hypothesis that the agent must verify

## B.2 Triplet Definitions

Here we further define our terms "pre-condition" "action sequence" and "post-condition."

Consider a state $s$ (here we ignore the state/observation distinction for clarity of explanation, this would be an observation in our POMDP). We can consider a pre-condition or post-condition to be a boolean function of the state $s$. Given a state $s$, we can say whether a particular pre-condition or post-condition is true or false. For example, the condition "the door is open" is a function of the state $s$, which we can write as $f_{thedoorisopen}(s)$ which is true when the door is open, and false when it is not. Action sequences are also boolean functions, but over a sequence of states and actions. So for instance, "the agent moves to the craftingtable" can be written as $f_{theagentmovestothecraftingtable}(s_1, s_2, ..., a_1, a_2, ...)$ for some sequence of states and actions, which is true when the agent starts not at the craftingtable and takes actions that move it to the craftingtable.

Now we can connect these three functions or conditions to causality. In this construction, what we are doing is hypothesizing a causal connection between the pre-condition and action sequence and the post-condition: $(\textit{pre-condition}, \textit{action sequence}) \implies \textit{post-condition}$. This is a logical statement, which can be true or false, which asserts that when we satisfy the pre-condition and take the action sequence, the resulting state will satisfy the post-condition.

Finally, we connect this discription of these terms to the string text of the actual hypotheses. We treated these three terms like exact logical functions, but we in fact translate these functions to text strings. So a logical function of whether the door is open becomes "the door is open." In this work, we write these strings ourselves, although in future work, these could be crowd-sourced, or generated as part of an interactive game. Human language is more complicated and less un-ambigious than mathematical formulations but as we note in the main paper, we train our agents to generalize to new hypotheses and ones that do not follow this exact structure. By our choice of using templated language, we keep the kinds of hypotheses agents can learn as general as possible. Future work might explore how to solve new hypotheses online from humans, or even generate its own hypotheses to solve.

In the next sub-section, we exhaustively list the triplet and non-triplet templates we use in this work.

## B.3 Templates

**Color Switch**:

**Pre-condition**:
if the COLOR switch is ON_OFF_SWITCHSTATE
when the COLOR switch is in the ON_OFF_SWITCHSTATE position
the COLOR switch is ON_OFF_SWITCHSTATE

**Action**:
""

**Post-condition**:
then the door is open
the door is passable
and we see the door is open
the door will open

**Finetune templates**:
the door can only be opened by switching the COLOR switch to ON_OFF_SWITCHSTATE
when we see the COLOR switch is ON_OFF_SWITCHSTATE the door must be open
if the COLOR switch turns ON_OFF_SWITCHSTATE the door opens
when we see the door open it must be that the COLOR switch is in the ON_OFF_SWITCHSTATE position
those who want to open the door must first switch the COLOR switch ON_OFF_SWITCHSTATE
no password just make the COLOR switch be ON_OFF_SWITCHSTATE to open the door
COLOR switch ON_OFF_SWITCHSTATE implies door is open
only the COLOR switch being ON_OFF_SWITCHSTATE opens the door
the door is open because COLOR switch is in the ON_OFF_SWITCHSTATE position

COLOR switch ON_OFF_SWITCHSTATE equals open door
the COLOR switch opens the door but only when it is ON_OFF_SWITCHSTATE
door is open must mean that COLOR switch is ON_OFF_SWITCHSTATE an ON_OFF_SWITCHSTATE means the door is open but only if it is COLOR
COLOR controls the door and it opens when it is ON_OFF_SWITCHSTATE
ON_OFF_SWITCHSTATE is the correct position of the COLOR switch and it opens the door
the switch that causes the door to be open when it is ON_OFF_SWITCHSTATE is COLOR
if you see COLOR switch then the door is open
the door is independent of the COLOR switch
if the door is not open then the COLOR switch must be ON_OFF_SWITCHSTATE
if the COLOR switch is not ON_OFF_SWITCHSTATE then the door is open
to make the door not open the COLOR switch must be not ON_OFF_SWITCHSTATE
whether the door is open is completely independent of the COLOR switch
the COLOR switch is what controls the door
a not ON_OFF_SWITCHSTATE COLOR switch opens the door

**Template Values**
**COLOR**:
blue
red
green
black

**ON_OFF_SWITCHSTATE**:
on
off

## Pushblock

**Pre-condition**:
whenever the pushblock is in the PUSHBLOCK_POSITION
if the pushblock is at the PUSHBLOCK_POSITION
the pushblock is at the PUSHBLOCK_POSITION

**Action**:
""

**Post-condition**:
then the door is open
the door is passable
and we see the door is open
the door will open

**SP_FULL_TRAIN**:
PUSHBLOCK_POSITION is the correct position for the pushblock for the door to open
if the door is open it must be that the pushblock is at the PUSHBLOCK_POSITION
when the door is open it is because the pushblock is in the PUSHBLOCK_POSITION
when the pushblock is at the PUSHBLOCK_POSITION the door is open
pushblock PUSHBLOCK_POSITION means door open
the door can only be opened when the pushblock is PUSHBLOCK_POSITION
if the pushblock is PUSHBLOCK_POSITION it means the door is open
PUSHBLOCK_POSITION pushblock opens the door
open door implies pushblock PUSHBLOCK_POSITION
open door means pushblock PUSHBLOCK_POSITION
door opens when PUSHBLOCK_POSITION is where the pushblock is
PUSHBLOCK_POSITION is the correct position for the pushblock to open the door
the door when the pushblock is PUSHBLOCK_POSITION is open
PUSHBLOCK_POSITION position of the pushblock causes the door to open
door only opens on PUSHBLOCK_POSITION pushblock
door can only open with pushblock being PUSHBLOCK_POSITION

the pushblock being at the PUSHBLOCK_POSITION is completely independent of the door
the pushblock being PUSHBLOCK_POSITION is independent of the door being open
the door state is independent of pushblock PUSHBLOCK_POSITION
PUSHBLOCK_POSITION pushblock and door are independent

**Pushblock values**:
**PUSHBLOCK_POSITION**:
left
right
top
bottom

## Crafting

**Pre-condition**:
when you are at LOCATION and you have CRAFTING_ITEM
you are at LOCATION and have in your inventory CRAFTING_ITEM
whenever you have a CRAFTING_ITEM and are at LOCATION

**Action**:
and you do CRAFTING_ACTION
then you CRAFTING_ACTION

**Post-condition**:
you now have CREATED_ITEM in your inventory
then CREATED_ITEM is created
and this creates CREATED_ITEM
so CREATED_ITEM is created and put in your inventory
then CREATED_ITEM is made

**Finetune Templates**:
to create a CREATED_ITEM you must have CRAFTING_ITEM and go to LOCATION and do the action CRAFT-ING_ACTION
CREATED_ITEM can be created by doing CRAFTING_ACTION at LOCATION when CRAFTING_ITEM is in inventory
whenever you do CRAFTING_ACTION and have CRAFTING_ITEM at LOCATION a CREATED_ITEM is made
you have CRAFTING_ITEM and go to LOCATION and CRAFTING_ACTION and CREATED_ITEM will be created
whoever does CRAFTING_ACTION at LOCATION with CRAFTING_ITEM gets CREATED_ITEM
if you have CRAFTING_ITEM at LOCATION and you CRAFTING_ACTION you get CREATED_ITEM
if you do CRAFTING_ACTION at LOCATION with CRAFTING_ITEM you make CREATED_ITEM
whenever you have CRAFTING_ITEM at LOCATION and do CRAFTING_ACTION then you make a CREATED_ITEM
having CRAFTING_ITEM in your inventory being at LOCATION and doing CRAFTING_ACTION creates CRE-ATED_ITEM
CREATED_ITEM can be made with CRAFTING_ITEM when you do CRAFTING_ACTION at LOCATION
CRAFTING_ITEM plus LOCATION plus CRAFTING_ACTION equals CREATED_ITEM
create a CREATED_ITEM by being at LOCATION with CRAFTING_ITEM and doing CRAFTING_ACTION
CRAFTING_ACTION at LOCATION creates CREATED_ITEM but only if you have a CRAFTING_ITEM
if you want to make a CREATED_ITEM then go to LOCATION with CRAFTING_ITEM and do CRAFTING_ACTION
CRAFTING_ITEM in inventory at LOCATION makes CREATED_ITEM if you do CRAFTING_ACTION
CREATED_ITEM when CRAFTING_ITEM at LOCATION and do CRAFTING_ACTION
if you are at LOCATION and do CRAFTING_ACTION you make CREATED_ITEM
if you are anywhere and do CRAFTING_ACTION with CRAFTING_ITEM you make a CREATED_ITEM
having CRAFTING_ITEM at LOCATION and doing CRAFTING_ACTION does not make a CREATED_ITEM
CREATED_ITEM is created by being at LOCATION and doing CRAFTING_ACTION
make a CREATED_ITEM by having a CRAFTING_ITEM and doing CRAFTING_ACTION
you have CRAFTING_ITEM and go to LOCATION and CRAFTING_ACTION and CREATED_ITEM will not be created
LOCATION plus CRAFTING_ACTION creates a CREATED_ITEM
with a CRAFTING_ITEM you can make a CREATED_ITEM by doing CRAFTING_ACTION

**Template Values**:
**CRAFTING_ITEM** :

iron
wood
stick
pickaxe
coal

**CREATED_ITEM**:
torch
bed

**LOCATION**:
craftingtable

**CRAFTING_ACTION**:
craft

## Carpole

**Pre-condition**:
when are in the COLOR zone
if you are in the COLOR
when the agent is within COLOR

**Action**:
""

**Post-condition**:
the gravity is now MULTIPLIER
gravity is MULTIPLIER
the wind pushes DIRECTION
there is a DIRECTION wind

**Finetune Templates**:
MULTIPLIER gravity is caused by the COLOR zone
if you go to COLOR zone then gravity is MULTIPLIER
the gravity is MULTIPLIER because the agent is in the COLOR
COLOR zone implies MULTIPLIER gravity
gravity is MULTIPLIER whenever you go into the a COLOR zone
the COLOR causes gravity to MULTIPLIER
to make gravity MULTIPLIER need to be in COLOR
COLOR zone gravity MULTIPLIER
a gravity multiplier of MULTIPLIER is caused by being in COLOR zone
COLOR equals wind DIRECTION
when the wind blows DIRECTION it is because you are in COLOR
COLOR zone causes DIRECTION wind
only being in COLOR makes the wind blow DIRECTION
DIRECTION wind is in the COLOR zone
when you are in the COLOR zone there is a DIRECTION wind
DIRECTION wind is caused by being in the COLOR zone
wind pushing DIRECTION whenever you are in COLOR
gravity is totally independent of COLOR
COLOR zone does not effect gravity it is independent
the wind is completely independent of the COLOR zone
independent of wind DIRECTION is COLOR
gravity is changed by being in COLOR but not MULTIPLIER
the effect of being in COLOR is opposite to gravity MULTIPLIER
the wind blows opposite of DIRECTION when in COLOR zone
being in COLOR causes the wind to blow opposite to DIRECTION

**Template Values**:
**COLOR**:
blue
red
green
black

**MULTIPLIER** :
decreased
increased

**DIRECTION**:
left
right

## C    LEARNING DETAILS AND HYPERPARAMETERS

One detail of the prediction network is that we need to keep a memory of past state sequences, hypotheses and ground truths so we can actually train our prediction network. We do this by simply keeping track of the last $N$ times our agent answered a question, and keeping these in a FIFO memory. When we update our prediction network, we randomly sample from this pool. This also necessitates a 100k step break in period to collect enough examples.

In our policy finetuning experiments, we also stabilize our dual optimization problem by trading of optimization of the policy network and the prediction network. We must also start with the prediction network so that the reward for answering correctly is meaningful.

No extensive grid search of hyper-parameters was conducted. At various stages we experimented with different hyperparameter values especially PPO parameters (timesteps per batch), learning rates, PPO epochs, and optimizer.

Experiments were performed on Tesla K40, TitanX and TitanX Pascals GPUs on 2-4 GPU workstations. Several jobs could be run on a single GPU with the greatest performance bottleneck being CPU cycles rather than GPU cycles or memory.

Table 2: Pretraining Hyperparameters

| Parameter | Value |
|---|---|
| Algorithm | PPO Schulman et al. (2017) |
| Timesteps per batch | 2048 |
| Clip param | 0.2 |
| Entropy coeff | 0.1 |
| Number of parallel processes | 8 |
| Optimizer epochs per iteration | 4 |
| Optimizer step size | $2.5e^{-4}$ |
| Optimizer batch size | 32 |
| Discount $\gamma$ | 0.99 |
| GAE $\lambda$ | 0.95 |
| learning rate schedule | constant |
| Optimizer | ADAM Kingma & Ba (2014) |
| Past Frame Window Size | 5 |

Table 3: Finetuning Hyperparameters

| Parameter | Value |
|---|---|
| Algorithm | PPO Schulman et al. (2017) |
| Timesteps per batch | 2048 |
| Entropy coeff | 0.1 |
| Number of parallel processes | 8 |
| Optimizer epochs per iteration | 4 |
| Optimizer step size | $1e^{-5}$ |
| Optimizer batch size | 32 |
| Discount $\gamma$ | 0.99 |
| GAE $\lambda$ | 0.95 |
| learning rate schedule | constant |
| Optimizer | SGD |
| Past Frame Window Size | 5 |

Basis of RL implementations was from Kostrikov (2018)

Table 4: Prediction Hyperparameters

| Parameter | Value |
|---|---|
| Timesteps per batch | 2048 |
| Optimizer step size | $1e^{-3}$ |
| Optimizer batch size | 128 |
| learning rate schedule | constant |
| Optimizer | ADAM Kingma & Ba (2014) |
| Memory Burn-in | 100000 |
| Memory Size | 200 |
| Alternate Training Window | 10000000 |

# D    NETWORK DETAILS

Although other works such as Chaplot et al. (2018) have investigated language-conditioned RL (usually in the form of instruction following), our hypothesis conditioned problem proved to be challenging, and required some novelty in network architectures. Figure 7 shows our network diagrams.

Other works such as Chaplot et al. (2018) have incorporated gated mechanisms between language and perception. Manchin et al. (2019) employs self-attention mechanism within convolutional layers and Choi et al. (2017) also employs a self-attention mechanism in a DQN. Neither work incorporates language and the architectures are quite different from each other. Figure 7 shows the policy and transformer architectures (this is also in the main text).

For the policy network, it was important to use key-value attention. That is: the hypothesis is fed into a seq2vec model and is used as the *key* of a dot-product attention mechanism. The state (the grid locations of the items in the world and the inventory of the agent if applicable) is fed as input to $N$ parallel MLPs. The output of the MLPs are then fed as the *values* of the attention mechanism. The output of the module is then fed into the final hidden layer of the actor-critic network.

For the prediction network, we use the popular transformer architecture Vaswani et al. (2017). Our prediction network encodes both the hypothesis and past observations (after they are passed through a one layer network) using transformer encoders. These sequences are then combined using a transformer to generate a final hidden state as output which is then fed to a final prediction layer and sigmoid function to get our binary prediction.

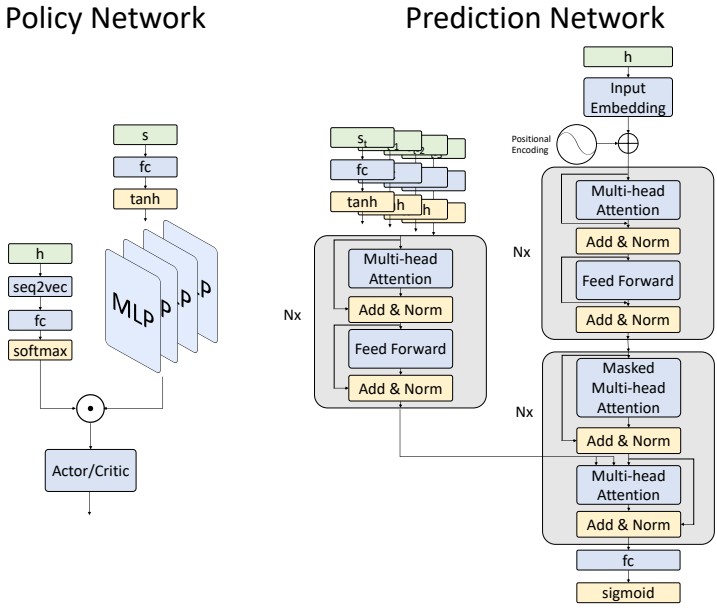

Figure 7: Network architecture for our policy network (left) and prediction network (right)

## D.1    NETWORK ABLATION

In Figure 8 we see the results of our network architecture ablation. As we can see, our new policy architecture described previosuly clearly outperforms a standard MLP policy network on the language-condition pretraining task. We also see that the transformer architecture outperforms the LSTM and MLP model on the final task when we hold the policy network constant.

## D.2    IMPLEMENTATION AND HYPERPARAMETERS

We take much of our implementation of transformers from Rush (2018).

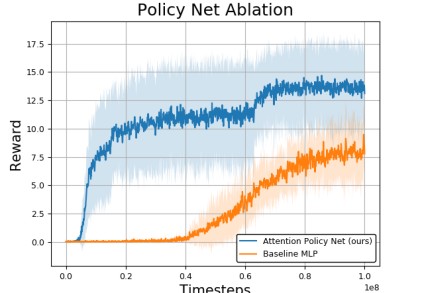 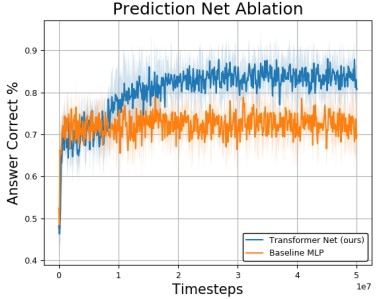

Figure 8: (left) policy network ablation (right) prediction network ablation.

Table 5: Policy Network Hyperparameters

| Parameter | Value |
|---|---|
| Seq2Vec Model | Bag-of-Words |
| Word Embedding Size | 32 |
| Hidden Size | 32 |
| MLP Num Hidden Layers | 2 |
| Number of MLP Modules | 16 |
| Transfer Layer | $tanh$ |

Table 6: MLP Baseline Policy Network Hyperparameters

| Parameter | Value |
|---|---|
| Seq2Vec Model | Bag-of-Words |
| Word Embedding Size | 32 |
| Hidden Size | 32 |
| MLP Num Hidden Layers | 2 |
| Transfer Layer | $tanh$ |

Table 7: Transformer Network Hyperparameters

| Parameter | Value |
|---|---|
| Word Embedding Size | 32 |
| Hidden Size | 32 |
| Transfer Layer | $ReLU$ |
| Transformer $N$ | 3 |

Table 8: Baseline Prediction Network Hyperparameters

| Parameter | Value |
|---|---|
| Seq2Vec Model | LSTM |
| LSTM Num Layers | 1 |
| Word Embedding Size | 32 |
| Hidden Size | 32 |
| MLP Num Hidden Layers | 2 |
| Transfer Layer | $tanh$ |

# E    TEST-TIME HYPOTHESES

To examine whether the hypothesis prediction networks are overfit to the very specific hypotheses the networks were trained on, we conduct a simple experiment and we test on different hypotheses. Here we keep the same underlying distribution of ground truth rules, but we change the hypotheses to ones not seen in any stage of training, and do not adapt the networks in any way to these hypotheses. The results are shown in Table 9 for the triplet pretraining method. We can see that for the most part, there is little to no drop in performance, with the exception of a substantial drop in crafting, but only for the fixed policy. The finetuned crafting policy still achieves comparable results to hypotheses it was trained on.

Table 9: Average Hypothesis Prediction scores on new templates

|  | **Method** | **Hypothesis Accuracy** |
|---|---|---|
| **ColorSwitch** | Fixed Policy | 81.49% |
| | Finetuned Policy | 71% |
| **Pushblock** | Fixed Policy | 89% |
| | Finetuned Policy | 84.87% |
| **Crafting** | Fixed Policy | 58.22% |
| | Finetuned Policy | 90.3% |
| **Cartpole** | Fixed Policy | 78.97% |
| | Finetuned Policy | 81.3% |

# F    CROSSTAB PERFORMANCE ANALYSIS

In this section, we further break down the results in the main paper by specific templates, entities mentioned (e.g. colors) and whether the given template was true or false. We do this analysis only for Fixed and Finetune as the baselines fail fairly uniformly and the analysis would be less interesting.

We get these results by running our pretrained models on at least 1000 of each subcategory listed and averaging the results across results (and also across runs as in our main results). This gives us a good sample for analysis of which types of hypotheses methods perform well or poorly on.

First we look at the accuracy sorting by whether the agent was given a true hypothesis to validate or a false hypothesis to invalidate in Tables 10,11,12, and 13. We see that, except for finetune accuracy on crafting, the agents have a generally much easier time confirming true hypotheses.

One possible explanation for this of course is that the observations that confirm a true hypothesis and the policy needed to do this are much simpler. To confirm a true hypothesis, the agent must essentially perform the hypothesis described exactly. So if the hypothesis is that flipping the green switch on opens the door, the correct policy is exactly going to the green door and toggling. And then the state affect is immediate. If following this causes the door to open, the state changes in a very obvious way - the door opens. Whereas if the statement was false, the agent would have to look at the sequence of states and infer that indeed did exactly the right hypothesis and the effect did not happen. Noticing a change in state is often easier than noticing a lack of change.

However, there may be other subtle things going on related to the agent behavior. For instance, if an agent's policy was to guess True if it does not know the ground truth, it would by definition do better on true hypotheses. Indeed, if we look at the actions pushblock agent, we notice this. Recall that if the block is against a wall, the agent may not be able to trigger the post-condition. But actually, the agent's odds here are better than random. If the block is against a wall, it can test two possible door conditions (if it's stuck on the left, it can test up and down). By doing this, it is able to test two alternate hypotheses. If the agent wants to test a hypothesis that moving to the right opens the door, it can test moving it to up and down. If it tries both of these and the door does not open, it is more likely that it is true than not. So the agent will guess True. This leads to a high True answer rate, but this stragegy will always fail in those less likely cases where that hypothesis was actually false.

Next we look at the "elements" i.e. the values in the templates such as the color of the switch or the item being crafted. See Tables 14,15,16, and 17. With the notable exception of Cartpole, the element

Table 10: Colorswitch T/F Crosstabs

| True/False | Fixed Acc | Finetune Acc |
|------------|-----------|--------------|
| True | **0.941558** | 0.835376 |
| False | **0.788904** | 0.721134 |

Table 11: Pushblock T/F Crosstabs

| True/False | Fixed Acc | Finetune Acc |
|------------|-----------|--------------|
| True | **0.969079** | 0.959164 |
| False | **0.760620** | 0.746091 |

does not substatially change the value. The exception is Cartpole, but this is actually because unlike the other environments, there are really two distinct hypothesis effects which are not equivalent: wind versus gravity. In the hypotheses which mention left or right, the agent is actually testing hypotheses about which way the wind is blowing (left of right) whereas the half and double hypotheses are testing the gravity. We see that the agent performs better on wind hypotheses than on gravity, but the direction or gravity effect does not substantially effect the result. This is likely due to the fact that when the pole of the cartpole is level, the effect of gravity is less obvious. If you read the original cartpole code, you can see this. So these are harder because it requires the agent to have the pole in the less stable, non-centered position. Whereas for wind, the effect of a wind force is immediate and more obvious to the agent (there is now an opposing force the agent must act against to keep the pole upright).

Finally, we break out results by each template in Tables 18,19,20, 21 and **??**. The results can be somewhat overwhelming, but a few general trends emerge. First, we can see that there is quite a bit of differing in performance for specific template types. Some specific templates can have very low performance for both fixed and finetuned. One very interesting result is for the special case templates. For Colorswitch and Pushblock, we find again that the general trend is that the fixed accuracy models perform a bit better (see Appendix G.3). But on Crafting and Cartpole, we see that finetuned are able to do significantly better than fixed. In Crafting this includes negation and recipes that are missing the location or input crafting item. In Cartpole this is negation and independence statements.

Table 12: Crafting T/F Crosstabs

| True/False | Fixed Acc | Finetune Acc |
|---|---|---|
| True | 0.819272 | **0.897819** |
| False | 0.730525 | **0.913886** |

Table 13: Cartpole T/F Crosstabs

| True/False | Fixed Acc | Finetune Acc |
|---|---|---|
| True | 0.876085 | **0.894504** |
| False | 0.822460 | **0.850877** |

Table 14: Colorswitch Elements Crosstabs

| Element | Fixed Acc | Finetune Acc |
|---|---|---|
| blue | **0.860603** | 0.817654 |
| black | **0.874475** | 0.711046 |
| red | **0.868067** | 0.699999 |
| green | 0.857772 | **0.883432** |
| on | **0.863618** | 0.779900 |
| off | **0.859275** | 0.772164 |

Table 15: Pushblock Elements Crosstabs

| Element | Fixed Acc | Finetune Acc |
|---|---|---|
| left | **0.881174** | 0.860691 |
| bottom | **0.869612** | 0.866311 |
| right | **0.833578** | 0.830409 |
| top | **0.874413** | 0.853540 |

Table 16: Crafting Elements Crosstabs

| Element | Fixed Acc | Finetune Acc |
|---|---|---|
| pickaxe | 0.749928 | **0.885686** |
| craft | 0.774918 | **0.905880** |
| torch | 0.790433 | **0.931534** |
| stick | 0.790664 | **0.927625** |
| coal | 0.766912 | **0.884030** |
| bed | 0.759223 | **0.880414** |
| iron | 0.783844 | **0.890114** |
| wood | 0.808903 | **0.939032** |

Table 17: Cartpole Elements Crosstabs

| Element | Fixed Acc | Finetune Acc |
|---------|-----------|--------------|
| blue    | 0.856136  | **0.882224** |
| black   | 0.844531  | **0.868902** |
| red     | 0.855209  | **0.898623** |
| green   | **0.841047** | 0.840754  |
| left    | 0.865710  | **0.876526** |
| right   | 0.873185  | **0.901369** |
| half    | 0.824625  | **0.846780** |
| double  | 0.839608  | **0.846751** |

Table 18: Colorswitch Template Crosstabs

| Template | Fixed Acc | Finetune Acc |
|----------|-----------|--------------|
| no password just make the COLOR switch be ON_OFF_SWITCHSTATE to open the door | **0.805164** | 0.715492 |
| the COLOR switch is ON_OFF_SWITCHSTATE then the door is open | **0.909462** | 0.809612 |
| COLOR controls the door and it opens when it is ON_OFF_SWITCHSTATE | **0.757894** | 0.636947 |
| those who want to open the door must first switch the COLOR switch ON_OFF_SWITCHSTATE | 0.738385 | **0.784663** |
| COLOR switch ON_OFF_SWITCHSTATE equals open door | **0.771802** | 0.654603 |
| if the COLOR switch is ON_OFF_SWITCHSTATE the door is passable | **0.909699** | 0.809354 |
| if the COLOR switch turns ON_OFF_SWITCHSTATE the door opens | **0.845888** | 0.808421 |
| the COLOR switch is ON_OFF_SWITCHSTATE the door is passable | **0.909085** | 0.806334 |
| when the COLOR switch is in the ON_OFF_SWITCHSTATE position the door is passable | **0.909311** | 0.798914 |
| the COLOR switch is ON_OFF_SWITCHSTATE and we see the door is open | **0.910084** | 0.808421 |
| ON_OFF_SWITCHSTATE is the correct position of the COLOR switch and it opens the door | **0.786958** | 0.617625 |
| if the COLOR switch is ON_OFF_SWITCHSTATE the door will open | **0.911169** | 0.813248 |
| when we see the COLOR switch is ON_OFF_SWITCHSTATE the door must be open | **0.860697** | 0.792323 |
| if you see COLOR switch then the door is open | **0.847181** | 0.761045 |
| when the COLOR switch is in the ON_OFF_SWITCHSTATE position then the door is open | **0.908866** | 0.805041 |
| the COLOR switch is what controls the door | **0.865501** | 0.790601 |
| when the COLOR switch is in the ON_OFF_SWITCHSTATE position the door will open | **0.911304** | 0.801073 |
| when the COLOR switch is in the ON_OFF_SWITCHSTATE position and we see the door is open | **0.909594** | 0.800795 |
| if the COLOR switch is ON_OFF_SWITCHSTATE and we see the door is open | **0.907675** | 0.805454 |
| the door is open because COLOR switch is in the ON_OFF_SWITCHSTATE position | **0.879004** | 0.792435 |
| the door can only be opened by switching the COLOR switch to ON_OFF_SWITCHSTATE | **0.741864** | 0.727885 |
| the switch that causes the door to be open when it is ON_OFF_SWITCHSTATE is COLOR | **0.766324** | 0.701296 |
| the COLOR switch is ON_OFF_SWITCHSTATE the door will open | **0.910904** | 0.809709 |
| if the COLOR switch is ON_OFF_SWITCHSTATE then the door is open | **0.909178** | 0.803808 |
| only the COLOR switch being ON_OFF_SWITCHSTATE opens the door | **0.800769** | 0.768345 |
| door is open must mean that COLOR switch is ON_OFF_SWITCHSTATE | **0.782568** | 0.749175 |
| an ON_OFF_SWITCHSTATE means the door is open but only if it is COLOR | 0.812291 | **0.813397** |
| when we see the door open it must be that the COLOR switch is in the ON_OFF_SWITCHSTATE position | **0.827992** | 0.757434 |
| the COLOR switch opens the door but only when it is ON_OFF_SWITCHSTATE | **0.842028** | 0.685985 |
| COLOR switch ON_OFF_SWITCHSTATE implies door is open | 0.804614 | **0.806208** |
| to make the door not open the COLOR switch must be not ON_OFF_SWITCHSTATE | **0.728761** | 0.698866 |
| if the door is not open then the COLOR switch must be ON_OFF_SWITCHSTATE | **0.855276** | 0.796445 |
| a not ON_OFF_SWITCHSTATE COLOR switch opens the door | **0.782377** | 0.704020 |
| if the COLOR switch is not ON_OFF_SWITCHSTATE then the door is open | **0.881558** | 0.805187 |
| whether the door is open is completely independent of the COLOR switch | **0.937662** | 0.850311 |
| the door is independent of the COLOR switch | **0.976774** | 0.803929 |

Table 19: Pushblock Template Crosstabs

| Template | Fixed Acc | Finetune Acc |
|---|---|---|
| door can only open with pushblock being PUSHBLOCK_POSITION | 0.820393 | **0.856789** |
| the pushblock is at the PUSHBLOCK_POSITION then the door is open | **0.869955** | 0.859727 |
| PUSHBLOCK_POSITION is the correct position for the pushblock to open the door | **0.872479** | 0.843071 |
| the pushblock is at the PUSHBLOCK_POSITION the door is passable | **0.872433** | 0.863410 |
| if the pushblock is PUSHBLOCK_POSITION it means the door is open | **0.876048** | 0.850846 |
| whenever the pushblock is in the PUSHBLOCK_POSITION the door will open | **0.874485** | 0.860658 |
| if the pushblock is at the PUSHBLOCK_POSITION the door is passable | **0.874790** | 0.859502 |
| PUSHBLOCK_POSITION pushblock opens the door | **0.878205** | 0.858935 |
| the pushblock is at the PUSHBLOCK_POSITION and we see the door is open | **0.870191** | 0.860378 |
| door opens when PUSHBLOCK_POSITION is where the pushblock is | **0.857907** | 0.853671 |
| if the door is open it must be that the pushblock is at the PUSHBLOCK_POSITION | **0.868565** | 0.840094 |
| if the pushblock is at the PUSHBLOCK_POSITION then the door is open | **0.877208** | 0.854677 |
| whenever the pushblock is in the PUSHBLOCK_POSITION the door is passable | **0.870555** | 0.866253 |
| open door means pushblock PUSHBLOCK_POSITION | 0.835129 | **0.857000** |
| if the pushblock is at the PUSHBLOCK_POSITION and we see the door is open | **0.871365** | 0.855871 |
| the door can only be opened when the pushblock is PUSHBLOCK_POSITION | **0.871308** | 0.855845 |
| pushblock PUSHBLOCK_POSITION means door open | 0.839570 | **0.861030** |
| whenever the pushblock is in the PUSHBLOCK_POSITION then the door is open | **0.878206** | 0.858112 |
| when the door is open it is because the pushblock is in the PUSHBLOCK_POSITION | **0.872247** | 0.846558 |
| whenever the pushblock is in the PUSHBLOCK_POSITION and we see the door is open | **0.870350** | 0.860942 |
| open door implies pushblock PUSHBLOCK_POSITION | 0.832304 | **0.861506** |
| if the pushblock is at the PUSHBLOCK_POSITION the door will open | **0.875567** | 0.856274 |
| when the pushblock is at the PUSHBLOCK_POSITION the door is open | **0.878546** | 0.847757 |
| the door when the pushblock is PUSHBLOCK_POSITION is open | **0.869970** | 0.849398 |
| the pushblock is at the PUSHBLOCK_POSITION the door will open | **0.872346** | 0.855193 |
| PUSHBLOCK_POSITION position of the pushblock causes the door to open | **0.871925** | 0.833536 |
| door only opens on PUSHBLOCK_POSITION pushblock | 0.833290 | **0.843573** |
| PUSHBLOCK_POSITION is the correct position for the pushblock for the door to open | **0.865061** | 0.846289 |
| the pushblock being PUSHBLOCK_POSITION is independent of the door being open | **0.845120** | 0.828433 |
| the door state is independent of pushblock PUSHBLOCK_POSITION | **0.848979** | 0.831997 |
| PUSHBLOCK_POSITION pushblock and door are independent | **0.838476** | 0.824016 |
| the pushblock being at the PUSHBLOCK_POSITION is completely independent of the door | **0.851769** | 0.836789 |

Table 20: Crafting Template Crosstabs

| Template | Fixed Acc | Finetune Acc |
|---|---|---|
| you are at LOCATION and have in your inventory CRAFTING_ITEM and you do CRAFTING_ACTION then CREATED_ITEM is made | 0.980752 | **0.992622** |
| if you want to make a CREATED_ITEM then go to LOCATION with CRAFTING_ITEM and do CRAFTING_ACTION | 0.618157 | **0.751656** |
| CREATED_ITEM when CRAFTING_ITEM at LOCATION and do CRAFTING_ACTION | 0.621186 | **0.833183** |
| you are at LOCATION and have in your inventory CRAFTING_ITEM then you CRAFTING_ACTION so CREATED_ITEM is created and put in your inventory | 0.909046 | **0.994463** |
| when you are at LOCATION and you have CRAFTING_ITEM then you CRAFTING_ACTION then CREATED_ITEM is made | 0.882973 | **0.990093** |
| when you are at LOCATION and you have CRAFTING_ITEM and you do CRAFTING_ACTION then CREATED_ITEM is made | 0.988334 | **0.992137** |
| if you do CRAFTING_ACTION at LOCATION with CRAFTING_ITEM you make CREATED_ITEM | 0.702412 | **0.981659** |
| you are at LOCATION and have in your inventory CRAFTING_ITEM and you do CRAFTING_ACTION you now have CREATED_ITEM in your inventory | 0.903610 | **0.993235** |
| whenever you have a CRAFTING_ITEM and are at LOCATION then you CRAFTING_ACTION you now have CREATED_ITEM in your inventory | 0.920456 | **0.991195** |
| CRAFTING_ITEM plus LOCATION plus CRAFTING_ACTION equals CREATED_ITEM | 0.493720 | **0.596603** |
| CREATED_ITEM can be created by doing CRAFTING_ACTION at LOCATION when CRAFTING_ITEM is in inventory | 0.431615 | **0.546072** |
| you are at LOCATION and have in your inventory CRAFTING_ITEM then you CRAFTING_ACTION then CREATED_ITEM is created | 0.980229 | **0.991235** |
| you are at LOCATION and have in your inventory CRAFTING_ITEM and you do CRAFTING_ACTION then CREATED_ITEM is created | 0.957643 | **0.993386** |
| CRAFTING_ACTION at LOCATION creates CREATED_ITEM but only if you have a CRAFTING_ITEM | 0.571485 | **0.913526** |
| when you are at LOCATION and you have CRAFTING_ITEM and you do CRAFTING_ACTION then CREATED_ITEM is created | 0.988463 | **0.994632** |
| whenever you have a CRAFTING_ITEM and are at LOCATION then you CRAFTING_ACTION and this creates CREATED_ITEM | 0.863886 | **0.990727** |
| whenever you have CRAFTING_ITEM at LOCATION and do CRAFTING_ACTION then you make a CREATED_ITEM | 0.859223 | **0.995383** |
| when you are at LOCATION and you have CRAFTING_ITEM and you do CRAFTING_ACTION and this creates CREATED_ITEM | 0.896466 | **0.993168** |
| to create a CREATED_ITEM you must have CRAFTING_ITEM and go to LOCATION and do the action CRAFTING_ACTION | 0.580112 | **0.747054** |
| whenever you have a CRAFTING_ITEM and are at LOCATION and you do CRAFTING_ACTION so CREATED_ITEM is created and put in your inventory | **0.868949** | 0.795353 |
| you are at LOCATION and have in your inventory CRAFTING_ITEM then you CRAFTING_ACTION then CREATED_ITEM is made | 0.935798 | **0.991869** |
| whenever you do CRAFTING_ACTION and have CRAFTING_ITEM at LOCATION a CREATED_ITEM is made | 0.655783 | **0.918542** |
| you are at LOCATION and have in your inventory CRAFTING_ITEM then you CRAFTING_ACTION and this creates CREATED_ITEM | 0.920180 | **0.994741** |
| whenever you have a CRAFTING_ITEM and are at LOCATION then you CRAFTING_ACTION then CREATED_ITEM is created | 0.963260 | **0.993860** |

Table 21: Crafting Template Crosstabs (Continued)

| Template | Fixed Acc | Finetune Acc |
|---|---|---|
| whenever you have a CRAFTING_ITEM and are at LOCATION and you do CRAFTING_ACTION and this creates CREATED_ITEM | 0.838137 | **0.994172** |
| whenever you have a CRAFTING_ITEM and are at LOCATION and you do CRAFTING_ACTION then CREATED_ITEM is created | 0.891907 | **0.914471** |
| you are at LOCATION and have in your inventory CRAFTING_ITEM and you do CRAFTING_ACTION and this creates CREATED_ITEM | 0.937222 | **0.990594** |
| you are at LOCATION and have in your inventory CRAFTING_ITEM then you CRAFTING_ACTION you now have CREATED_ITEM in your inventory | 0.885896 | **0.989596** |
| having CRAFTING_ITEM in your inventory being at LOCATION and doing CRAFTING_ACTION creates CREATED_ITEM | 0.487408 | **0.817408** |
| create a CREATED_ITEM by being at LOCATION with CRAFTING_ITEM and doing CRAFTING_ACTION | 0.545770 | **0.919413** |
| when you are at LOCATION and you have CRAFTING_ITEM and you do CRAFTING_ACTION so CREATED_ITEM is created and put in your inventory | 0.956038 | **0.991977** |
| whoever does CRAFTING_ACTION at LOCATION with CRAFTING_ITEM gets CREATED_ITEM | 0.483942 | **0.566790** |
| when you are at LOCATION and you have CRAFTING_ITEM then you CRAFTING_ACTION so CREATED_ITEM is created and put in your inventory | 0.973726 | **0.996132** |
| you have CRAFTING_ITEM and go to LOCATION and CRAFTING_ACTION and CREATED_ITEM will be created | 0.703044 | **0.914437** |
| CRAFTING_ITEM in inventory at LOCATION makes CREATED_ITEM if you do CRAFTING_ACTION | 0.545541 | **0.897572** |
| you are at LOCATION and have in your inventory CRAFTING_ITEM and you do CRAFTING_ACTION so CREATED_ITEM is created and put in your inventory | 0.898226 | **0.919370** |
| LOCATION plus CRAFTING_ACTION creates a CREATED_ITEM | 0.541849 | **0.873599** |
| when you are at LOCATION and you have CRAFTING_ITEM and you do CRAFTING_ACTION you now have CREATED_ITEM in your inventory | 0.972194 | **0.994262** |
| when you are at LOCATION and you have CRAFTING_ITEM then you CRAFTING_ACTION and this creates CREATED_ITEM | 0.941614 | **0.994693** |
| CREATED_ITEM can be made with CRAFTING_ITEM when you do CRAFTING_ACTION at LOCATION | 0.578862 | **0.687136** |
| whenever you have a CRAFTING_ITEM and are at LOCATION and you do CRAFTING_ACTION then CREATED_ITEM is made | 0.891969 | **0.993853** |
| whenever you have a CRAFTING_ITEM and are at LOCATION then you CRAFTING_ACTION then CREATED_ITEM is made | 0.933055 | **0.989965** |
| whenever you have a CRAFTING_ITEM and are at LOCATION then you CRAFTING_ACTION so CREATED_ITEM is created and put in your inventory | 0.900469 | **0.923228** |
| whenever you have a CRAFTING_ITEM and are at LOCATION and you do CRAFTING_ACTION you now have CREATED_ITEM in your inventory | 0.909898 | **0.993880** |
| when you are at LOCATION and you have CRAFTING_ITEM then you CRAFTING_ACTION you now have CREATED_ITEM in your inventory | 0.938361 | **0.993092** |
| if you have CRAFTING_ITEM at LOCATION and you CRAFTING_ACTION you get CREATED_ITEM | 0.832800 | **0.993352** |
| when you are at LOCATION and you have CRAFTING_ITEM then you CRAFTING_ACTION then CREATED_ITEM is created | 0.885430 | **0.992530** |
| you have CRAFTING_ITEM and go to LOCATION and CRAFTING_ACTION and CREATED_ITEM will not be created | 0.777171 | **0.827805** |
| having CRAFTING_ITEM at LOCATION and doing CRAFTING_ACTION does not make a CREATED_ITEM | 0.647106 | **0.800454** |
| make a CREATED_ITEM by having a CRAFTING_ITEM and doing CRAFTING_ACTION | 0.531967 | **0.757789** |
| if you are anywhere and do CRAFTING_ACTION with CRAFTING_ITEM you make a CREATED_ITEM | 0.698206 | **0.875884** |
| with a CRAFTING_ITEM you can make a CREATED_ITEM by doing CRAFTING_ACTION | 0.550868 | **0.885985** |
| if you are at LOCATION and do CRAFTING_ACTION you make CREATED_ITEM | 0.872249 | **0.999038** |
| CREATED_ITEM is created by being at LOCATION and doing CRAFTING_ACTION | 0.676661 | **0.872774** |

## G  ADDITIONAL DISCUSSION / DETAILS

### G.1  PRETRAINING ADDITIONAL DISCUSSION

For the ColorSwitch environment, we found that pretraining with just the pre-condition reward leads to better results for the Color Switch environment and show those results here. We chose $C = 10$ for this proxy reward, so we see from our figure that we are able to achieve the pre-condition toggling almost perfectly.

For Pushblock, we use both $R_{\text{pre+post}}$ and $R_{\text{post}}$, however, because of the afformentioned issue of the block getting stuck against a wall, it does not achieve perfect success each time.

For Crafting we see that pretrain results converge towards around 15. This is because the best the pretraining can do is to try to complete the recipe. If the hypothesis is false, however, we often cannot toggle the post-condition because it often requires more than $K = 5$ steps to complete seperately from toggling the pre-condition.

Similarly for Cartpole, we see the same thing, except it can get to around 17.5. This is because if the true and false zones are adjacent to each other, the agent can toggle both the pre- and post-conditions successfully (try pre-condition, when it doesn't work, move the the adjacent zone where the effect is true).

### G.2  ALTERNATIVE CUTOFFS FOR ADAPTATION TRAINING

As promised, in Figure 9 we show the adaptation results when we choose an 80% cutoff instead of 90%.

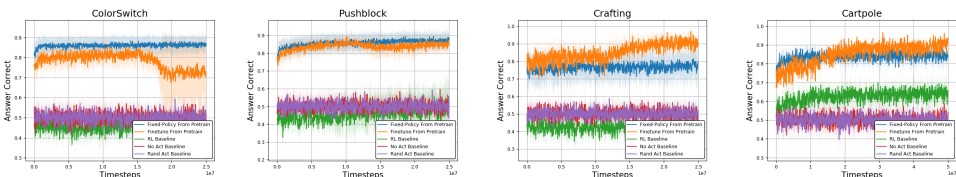

Figure 9: Hypothesis prediction accuracy on *both* triplet *and* non-triplet hypotheses for the Color Switch, Pushblock, Crafting and Cartpole environments, using $R_{\text{Hyp}}$ reward for training.

### G.3  TRAINING VARIANCE FOR PUSHBLOCK AND COLORSWITCH

In the main paper we discussed the problem of the finetuning training adding additional variance to training, resulting in worse average performance. First, we can actually see this from the training curves themselves in Figure 4 in the main text. The orange curves of the pretraining reward matches the blue fixed reward curve, but has more variance. In the ColorSwitch example, it has a hickup around $1.5e7$ timesteps that it does not fully recover from. This is because, unlike the fixed method, this method has to simultaneously train both the policy network and the prediction network. So when the policy network achieves a less-optimal policy, it becomes more difficult to train the prediction network which feeds back into the policy network. Whereas the fixed method only trains prediction and is much more stable during training.

We also mention that for some runs, this becomes less extreme. Some runs of the finetune policy do much better than average. In colorswitch, the best finetune run achieves $83.5\%$ accuracy versus $90.5\%$ for fixed and for pushblock achieves $87.6\%$ versus $89\%$, a smaller difference than the averages in Table 1.

# H   INTRINSIC PRE-TRAINING EXPERIMENTS

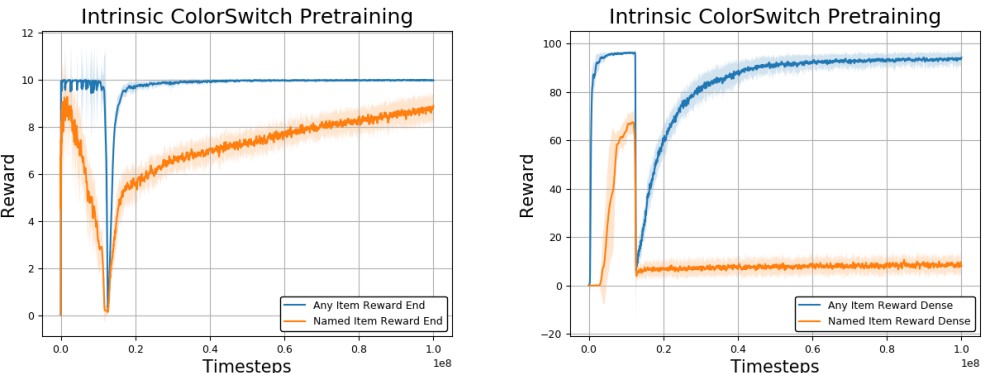

Figure 10: Pretraining Reward for ColorSwitch for intrinsic motivation. Showing mean and variance bands on 25 random seeds.

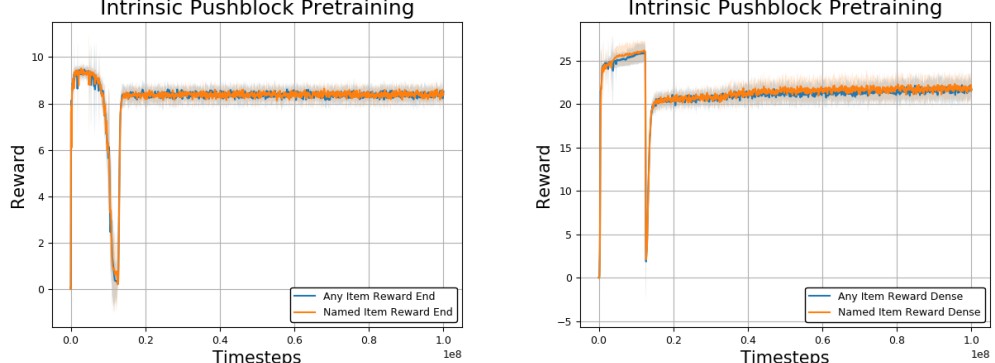

Figure 11: Pretraining Reward for Pushblock for intrinsic motivation. Showing mean and variance bands on 25 random seeds.

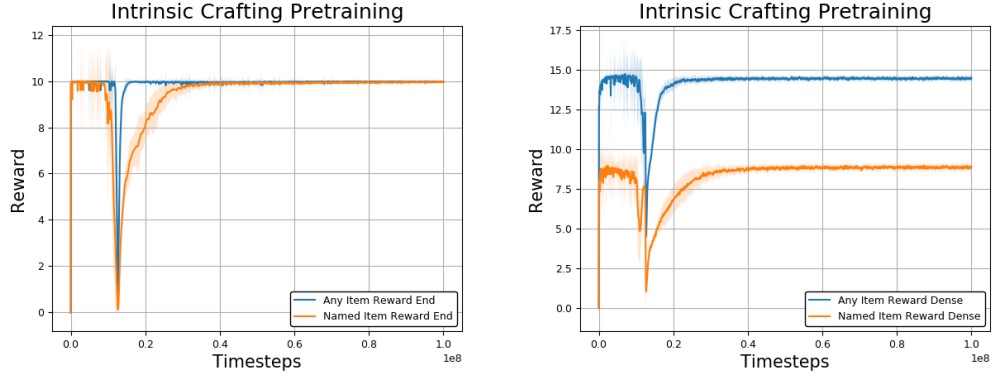

Figure 12: Pretraining Reward for Crafting for intrinsic motivation. Showing mean and variance bands on 25 random seeds.

In this section we expand on the experiments from section 6 of the main paper.

In this experiment, we show results on our hypotheses verification problem using different forms of intrinsic motivation pre-training. We show results for 4 different pretraining schemes:

1. Change any item state in the world. Receive reward at end.
2. Change any item referenced in the hypothesis. Receive reward at end.
3. Change any item state in the world. Receive reward instantaneously.
4. Change any item referenced in the hypothesis. Receive reward instantaneously.

Reward at the end means that it operates similar to hypothesis pre-training. Specifically, the agent get reward only at the end of the episode when it has taken a stop action. At that step it gets a $+C$ reward if it changed within the last $N$ frames. For these rewards, we choose $C = 10$. Instantaneous reward is what it sounds like. When the object state is changed, the reward is instantly received by the agent. We chose $C = 1$ for colorswitch and pushblock and $C = 5$ for crafting. Item means any object that is not the agent. So this includes crafting items, switches, pushblocks, etc. We show results on 25 random seeds. We preserve all training and network hyper-parameters.

We show the rest of the training curves for our intrinsic experiments in Figure 10, 11, 12 we show the pretraining curves for our intrinsic rewards.

All training and network parameters are kept the same from earlier experiments.

