# OpenReview forum: "Empirically Verifying Hypotheses Using Reinforcement Learning"
_ICLR.cc/2021/Conference — Reject_

### Official Review · AnonReviewer2 · 2020-10-26

**Rating:** 3
**Confidence:** 4

**Review:**


This paper presents the problem formulation for a hypothesis testing within a reinforcement learning paradigm. In particular, given a hypothesis represented as a formal, templated language, an agent needs to perform a sequence of actions and check whether the hypothesis holds true or false.

The paper proposes multiple methods since it is shown that traditional RL is insufficient for an agent to learn to test hypotheses. One of methods is to use pre-training taking advantage of more detailed information (represented as a triplet of pre-condition, action sequences, and post condition) about the hypothesis verification. Authors further investigated training after a pretraining step, and hypotheses that cannot be represented as a triplet.

Hypothesis verification using RL framework seems an important research direction. Based on the experiments presented in this paper, my understanding about the hypothesis testing is that it is about performing a complex task relevant to the hypothesis, and answering whether the hypothesis turns out to be true or false. Providing a triplet of information seems to be similar to the idea of breaking up such a complex task into comprehensible smaller tasks (like goal vs. subgoals) since the way pre-training works in this paper is precisely nudging the agent to perform subtasks. In this regards, I am not quite convinced the novelty of the formulation or methods.

In MineRL competition 2020, for example, the goal of the competition is to mine diamond. Agents receive rewards for obtaining items which are necessary to mine diamond. Hence, one may think that the agent needs to implicitly generate hypotheses and test the hypotheses to finally obtain diamond. This seems a harder task than hypothesis verification tasks this paper considered. How about "Montezuma's Revenge"?

Unfortunately, I found this paper paper does not adequately provide neither novel methods, results, nor insights. I would like to reexamine the paper once the authors respond focusing on which parts are particularly novel.

Minor points.
Figure 2. "the the" for color switch
Page 23, missing reference "??"
One switches (pg 5, color switch)

Question
"RL agents struggle to solve the problem." Is this because of fundamental flaws in RL formulation (sparse reward, credit assignment problem, etc.) or just not having enough time steps (say, 1e9, 1e10 ...). Without having experiments on a simple hypotheses and smaller grid-world, it is difficult to examine the performance of standard RL agents.

---

> ### Author Response · Authors · 2020-11-18
> **Thank you**
>
> We thank the reviewer for their time and feedback

---

### Official Review · AnonReviewer1 · 2020-10-26
**ill-formalized problem**

**Rating:** 3
**Confidence:** 3

**Review:**

The paper "Empirically Verifying Hypotheses Using Reinforcement Learning " proposes learning strategies of agents where goals are to validate or invalidate hypotheses about the world. This is done via RL, using the structure of triplet {pre-condition; action-sequence; post-condition} to drive the learning with intrinsic rewards.

While the problem is interesting, I found the paper difficult to read as the task is ill-defined in the section 3 where many notation definitions are missing and some notations are reused in different contexts with different definitions (e.g.  o=sw,h in the first sentence of last paragraph of section 3 and o is an observation from the world in the beginning of section 4, a is {false,true} in section 3 but can also be a move in the world later). Also, the state includes the dynamics L_W ? I understand that there are two levels of actions, states and observations (moves in the world and final decision w.r.t. the hypothesis) but this presentation could be made much clearer for the reader.

More importantly, my main concern is about the experimental setting. In the setting formalized in section 3, I understood that we get hypotheses and ground truth about their validity in the sampled world. Ok, but if we got the ground truth, what the learning could be used for. The only setting that matters in that case is from my point of view a meta-rl setting where we train agents to decide from a training set of worlds (for which we know the ground truth of hypotheses), which we hope well generalize for other worlds with similar properties. However, I cannot see any discussion about this, neither in the formalization section nor after in the experiments. Reported experiments only give results at train time from what I understood, which is clearly not sufficient to demonstrate the benefits of the approach. How the strategies generalize, and in which conditions they do is of crucial matter here.

In the motivation from section 3, authors justify their approach by claiming that triplet hypotheses can be useful for interfacing symbolic solvers. It would also have been nice such kind of setting. Also, I do not understand the justification of natural language hypotheses (i.e., "scaling annotations to untrained humans").   It would have been nice to have experiments more related to these announced interests,  to highlight the usefulness of the approach.

At last, do we use text encoding for natural language hyoptheses described in the special cases setting ? Please detail.

Globally, I feel that the paper needs a full rewritting to improve clarity. And that experiments must include generalization results.

---

> ### Author Response · Authors · 2020-11-18
> **Thank you**
>
> We thank the reviewer for their time and feedback

---

### Official Review · AnonReviewer4 · 2020-10-30
**Interesting, but too much of reward engineering**

**Rating:** 5
**Confidence:** 3

**Review:**

# Summary
​
This paper introduces a problem setting where an RL agent must interact with its environment to predict whether a given hypothesis is true or false. On modified versions of environments like gridworld and cartpole, they show that PPO with a sparse reward is unable to correctly test the hypothesis. The key technical contribution is making the reward more dense by assuming a predefined structure in a subset of hypotheses. This denser reward is used to pretrain the RL policy, which is then finetuned over the rest of the hypotheses using sparse reward.
​
----------
​
# Strengths
​
- The introduced problem of testing various hypotheses about environments is a novel and interesting application of RL.
​
- The paper shows empirically that it is hard to solve this problem with only sparse rewards, even for simple environments. I especially appreciate the oracle ablations, which show that the main issues is of credit assignment - the joint problem of learning an exploring policy and hypothesis predictor together is hard, while the individual problems themselves were easily solvable.
​
- The paper is clearly written and nicely defines the hypothesis verification problem. It was easy to understand what the key problem, intuition, and contributions are.
​
----------
​
# Weaknesses
​
- The technical novelty of the paper is a way to engineer a denser reward based on priviledged information about the space of hypotheses. While the problem of testing hypothesis with environment interaction is interesting, one expects a more principled solution for this problem. Moreover, it is not convincing that such Triplet hypotheses would be available in practical scenarios where such hypothesis testing may be applied. If the authors want to claim exploiting triplet hypotheses as the key contribution, then they should add more realistic environments to justify their use.
​
- The method makes strong assumptions about what is available at training time, such as direct supervision for prediction network and the underlying (pre-condition, action, post-condition) structure of triplet hypotheses. It is fine to make such assumptions, but then the evaluation must be done for held-out environments as well as held-out hypotheses, where no reward information is available (because having a reward function means you already know the answer to the hypothesis). This would truly show that the trained agent can test new hypotheses in new environments (under the same distribution as training).
​
- The fact that intrinsic rewards can sometimes succeed at solving the task (Fig. 5 - crafting finetuning) suggests that perhaps it is not necessary to assume triplet hypotheses for solving these tasks. Potentially, more principled exploration approaches will automatically solve this problem (just like other sparse reward RL problems). Also, the reward curves in Figs 10 - 12 do not look normal. The training breaks down in the middle and then gets back up. Is it possible that the method is not properly trained and just requires more hyperparameter tuning and careful training? Or is there some other justification for such plots?
​
----------
​
# Reason for decision
​
While the introduced problem is interesting (building upon Denil et. al), my main concern for recommending weak reject is that the main contribution seems to be just problem-specific reward engineering. This can still be a valid contribution, but the assumptions would then need to be justified by evaluating in more realistic environments (like Denil et. al did with physics experiments). Even then, it is a philosophical debate whether reward engineering is the right way to approach the problem, or developing more principled exploration methods.
​
----------
​
# Suggestions for improvement
​
- Some examples of more challenging environments for hypothesis testing: continuous control robotic experiments (simulation) where objects need to be analyzed to test hypothesis; visual navigation tasks where a house needs to be explored to gather information.
​
- A lot of important information is missing from the main paper and is present in the Appendix. The main paper should include more information about the inputs (text), the structure of policy network (seq2vec), prediction network (transformer), and important training details (e.g. PPO being the RL algorithm).

---

> ### Author Response · Authors · 2020-11-18
> **Thank you**
>
> We thank the reviewer for their time and feedback

---

### Official Review · AnonReviewer3 · 2020-11-01
**Paper lacks careful exposition on an interesting and widely studied topic.**

**Rating:** 4
**Confidence:** 4

**Review:**

This paper considers the general problem of testing hypotheses about the world by a kind of reinforcement learning, much as a person might learn by taking actions and observing their outcomes -- equivalently learning policies that can generate observations to validate a hypothesis.   A hypothesis is a symbolic representation of precondition, action and effects. The paper takes advantage of the ability of reinforcement learning to manipulate the environment that makes it possible to make causal inferences about the world.

That said, I find the paper's inadequate explanation and lack of rigor in the formal definition of the problem an impediment to understanding the work. As well as I can make out, it posits a distribution over "worlds" and "hypotheses" h, where worlds are specified by rules L^w, but also as triples of S_w, A_w & T_w, where T_w depends on L_w, (and conceivable also on A_w). One posits a truth functional G(h, W)  (Where does this come from?), by which a reward is generated if G equals a policy \pi(O_t, h), O_t an observation sequence, that is also a truth functional whose action space is True or False.  In short, it looks like we are talking about an RL that makes a guess whether the current hypothesis applies in the current world state.

However, at this point things break down.  A new truth functional f(O_t, h) (or is this the prediction network, mentioned on p5?) is introduced, possibly to replace \pi(), and possibly driven by a "policy network" for purposes of "pre-training"? The exact formalization of a hypothesis, from it's textual expression is missing. So is the source of G(), which one might assume is an oracle by which policies are validated?

Furthermore the stated need to "predict the hypothesis" is a phrase that doesn't ring true. Strictly predictions apply to outcomes (a hypothesis and pre-condition predict an outcome), but a hypothesis would be inferred from outcomes. "Prediction" in this sense may simply be a colloquialism, just adding to the confusion.

Recourse to the literature discussion is to no avail. The references are vague or not clearly relevant.  Brachman & Levesque 2004 is an entire anthology so it's not clear what exactly is being referred to. The term "action sentence" that is referenced as Pearl 2009 does not even appear in the cited work. The citation to Mao 2019 is a weak analogy, and so on.

In summary, without clear definitions and careful adherence to notation, critical evaluation of the paper is encumbered.  Similarly, the domain being stochastic apparently, any discussion how probabilistic aspects are treated is lacking and needs to be addressed

---

> ### Author Response · Authors · 2020-11-18
> **Thank you**
>
> We thank the reviewer for their time and feedback

---

### Decision · Program_Chairs · 2021-01-07
**Final Decision**

**Decision:**

Reject

**Comment:**

All the reviewers unanimously agree that the paper should be rejected. The main concern is well summarized by comment by R1's comment "While the problem is interesting, I found the paper difficult to read as the task is ill-defined in section 3 where many notation definitions are missing and some notations are reused in different contexts with different definitions". Also, as R4 mentions the proposed method can be reduced to reward engineering and doesn't provide any scientific or methodological advancement to the problem of testing hypothesis. The authors did not provide any rebuttal.